## Original Research Article

Arabidopsis; curation; leaf anatomy; modelling; organelle.

**Corresponding author:**
Gilles Curien; Email: gilles.curien@cea.fr

**Associate Editor:**
Dr. George Bassel

# The Arabidopsis leaf quantitative atlas: a cellular and subcellular mapping through unified data integration

Dimitri Tolleter[1], Edward N. Smith[2] , Clémence Dupont-Thibert[1], Clarisse Uwizeye[1], Denis Vile[3], Pauline Gloaguen[1], Denis Falconet[1], Giovanni Finazzi[1], Yves Vandenbrouck[4] and Gilles Curien[1]

[1]Laboratoire de Physiologie Cellulaire et Végétale, Université Grenoble Alpes, CNRS, CEA, INRAE, Grenoble, France; [2]Molecular Systems Biology, Groningen Biomolecular Sciences and Biotechnology Institute, University of Groningen, Groningen, The Netherlands; [3]Laboratoire d'Ecophysiologie des Plantes sous Stress Environnementaux (LEPSE), UMR 759, Université de Montpellier, INRAE, Institut Agro, Montpellier, France; [4]CEA DRF, Gif-sur-Yvette, France

## Abstract

Quantitative analyses and models are required to connect a plant's cellular organisation with its metabolism. However, quantitative data are often scattered over multiple studies, and finding such data and converting them into useful information is time-consuming. Consequently, there is a need to centralise the available data and to highlight the remaining knowledge gaps. Here, we present a step-by-step approach to manually extract quantitative data from various information sources, and to unify the data format. First, data from Arabidopsis leaf were collated, checked for consistency and correctness and curated by cross-checking sources. Second, quantitative data were combined by applying calculation rules. They were then integrated into a unique comprehensive, referenced, modifiable and reusable data compendium representing an Arabidopsis reference leaf. This atlas contains the metrics of the 15 cell types found in leaves at the cellular and subcellular levels.

## 1. Introduction

Quantitative reasoning provides interesting insights into relations between biological entities, and is useful when seeking to support or reject hypotheses. However, modellers, biochemists and cell biologists often have difficulty collecting the required quantitative datasets. This is true even for well-characterised biological systems. For example, for the model plant *Arabidopsis thaliana*, many studies have been devoted to molecular or genetic analyses. Nevertheless, it is far from trivial to determine the metrics of the anatomy of this plant. Different imaging techniques (TEM, FIB-SEM, SEM, confocal microscopy etc.) target specific scales (from the micrometre to the centimetre), and used alone, none can provide a complete quantitative view of a leaf from the subcellular or sub-organelle level to the tissue level. In addition, organ plasticity in higher plants is considered a significant hurdle to providing 'reference' metrics. Indeed, all experimentalists are well aware of the quantitative variations in the size of plant organ or cell volumes as a function of growth conditions. However, the study by Massonnet et al. (2010) showed that when growth conditions are carefully controlled, plant macroscopic features are conserved between different laboratories. In any case, to go further into the characterisation of plants and reach a quantitative understanding of metabolism, reference states are needed. Even if these states are approximate, they allow for calculations. Here, we developed a quantitative atlas of Arabidopsis leaves. To construct this atlas, we primarily relied on a quantitative analysis of reference leaf 6 and its major cell types (mesophyll and epidermis) thanks to the pioneering work of Wuyts et al. (2010) and Wuyts et al. (2012) . Leaf 6 is the first adult leaf, previously used as a reference leaf in developmental and proteomics studies (Baerenfaller et al., 2012, 2016; Cookson et al., 2006; Granier et al., 2002; Seaton et al., 2018). Among the descriptive elements, we provide metrics for vessels and specialized cells in reference leaf 6. Subcellular metrics (organelle numbers, volumes and areas) were collated from the literature for the various cell types in the leaf, when available. Data used herein were published 'as is' and not supported by any Data Management Plan. As a consequence, data and numerical values had to be extracted manually from human readable sources (tables in papers, texts,

graphs and supplementary data) and carefully examined to make them accessible and reusable as a first step to fit with Findability, Accessibility, Interoperability, and Reuse (FAIR) principles (Wilkinson et al., 2016). Therefore, the way the data were organised to allow calculations was driven by the nature of the biological material and the kinds of data available rather than on any pre-established protocols. Formulas are included in an Excel file allowing data to be dynamically updated as new data become available. We provide illustrations of how this data compendium can be used to calculate membrane surface areas at the leaf level and to convert metabolite amounts in the various units used in the literature to a common unit, taking metabolite subcellular localisation into account. This quantitative atlas could be useful in many different contexts where quantitative parameters are required: for example, knowledge of cell/organelle volumes was essential for modelling in Beauvoit et al. (2014), Shameer et al. (2020) and Topfer et al. (2020), and knowledge of cell type and number was used in Scheunemann et al. (2018) and Hunt et al. (2023). Furthermore, this quantitative atlas can also feed into other integrative approaches such as the Plant Cell Atlas initiative (Fahlgren et al., 2023; Jha et al., 2021).

## 2. Results

### 2.1. Cellular metrics of Arabidopsis reference leaf 6

To build the quantitative atlas of an Arabidopsis leaf under reference growth conditions (Supplementary Table S1), we followed the rationale illustrated in Supplementary Figure S1. For the cellular and subcellular metrics, we favoured the most comprehensive articles and complemented missing data with information from other studies, performed under experimental conditions that were as close as possible to those of the reference study (Wuyts et al., 2012). We chose to map the expanded *Arabidopsis thaliana* Col-0 Leaf 6, 21 days after leaf initiation (stage 1.09 according to Boyes et al., 2001). In Wuyts et al. (2012), plants were grown under 'high cumulative light' (16 h light per 24 hour, 9.6 mol photons m$^{-2}$ day$^{-1}$, i.e., 166 µmol m$^{-2}$ s$^{-1}$), and well-watered conditions (see Supplementary Table S.1.1, lines #6 to #17). The quantitative analysis provided by Wuyts et al. (2012), based on multiphoton laser scanning microscopy analysis of Arabidopsis leaf, is the most comprehensive available so far. It encompasses five cell types of the expanded *Arabidopsis thaliana* Col-0 leaf 6: lower (abaxial) and upper (adaxial) epidermal cells, stomata, palisade and spongy mesophyll cells (see Figure 1 and Supplementary Table S1.1). We assume that the small fraction of the fully expanded leaf reconstructed by Wuyts et al. (2012) is representative of the rest of the leaf for the photosynthetic and epidermal cells. For the cell types not analysed in Wuyts et al. (2012) – e.g., bundle sheath cells, phloem and xylem cells and trichome cells – we extracted quantitative data from selected publications, where plants were grown under similar conditions and to a similar age, whenever possible (see Supplementary Figure S1). This included selection of studies using the long-day light regime. In some specific instances where data were unavailable, we had to consider data obtained with plants grown under short-day conditions (see the 'growth conditions' column in Supplementary Table S1). Cell densities, reference ratios such as leaf fresh weight/leaf dry weight (LFW/LDW) or the number of leaves per g LFW were derived and used in additional calculations (cell numbers, total cell volumes, etc.) as detailed in Supplementary Table S1.1 (and Table 1).

The specific geometry of veins in a leaf – with distinct anatomies for primary veins, secondary veins, and higher order veins – was accounted for in specific calculations, as detailed in Supplementary Table S1, sheets S1.2 and S1.3 (and Section 4). In addition to cell numbers and volumes, the data in Supplementary Table S1 include experimental errors, calculations (in bold), references to the original studies, comments, methods used, growth conditions and plant growth stage. To reflect data quality (difference in experimental conditions, data reliability, hypothesis or extrapolations), Excel cells in Table S1 were colour coded (see Supplementary Figure S1 and Supplementary Table S1, sheet S1.10, lines #4, #6 and #8).

A summary of cellular metrics is provided in Supplementary Table S1, sheet S1.7 (data from Supplementary Table S1, sheet S1.7 are displayed in Table 2 and Figure 2).

When data on the anatomy of mature Arabidopsis leaf 6 were integrated, it emerged that vein cells were nearly as abundant as photosynthetic cells (286,000 veins cells, 295,000 bona fide photosynthetic cells and 183,000 epidermal cells; Table 2 and Figure 2a). We were unable to confirm these cell counts based on the literature. Nevertheless, these conclusions appear plausible. The high abundance of vein cells is linked to the length of the veins (37 cm in reference leaf 6 with a surface of 121 mm$^2$; see Supplementary Table S1.2, line #19) and the very small size of individual vein cells. Indeed, a sieve element is 700 times smaller than a palisade mesophyll cell (see Table 2, and Supplementary Table S1.2, line #143 and Supplementary Table S1.1, lines #72). The total number of cells in reference leaf 6 (121 mm$^2$; Wuyts et al., 2012) was estimated to be ~764,000 cells (Table 2). For a 3-week-old *Arabidopsis thaliana* Col-0 rosette with 29 leaves, and a total leaf area of 10,500 mm$^2$ (i.e., ~1 dm$^2$; see Figure 2 in Aguirrezabal et al., 2006), we can thus extrapolate a total of 66.3 million cells.

Volumetric data provided a different picture (see Table 2 and Figure 2b). As expected, the volumetric contribution of *bona fide* photosynthetic cells (mesophyll and bundle sheath cells) dominated leaf volume (86%). Among photosynthetic cells, the greatest volume is occupied by palisade mesophyll cells (75%), followed by spongy mesophyll cells (21%). Epidermal cells represent 14% of the total cellular volume. In contrast, bundle sheath cells, surrounding the secondary and higher-order veins, represent only 3% of the total volume of photosynthetic cells. Despite their functional importance, vein cells (excluding bundle sheath cells) contribute only 0.7% of the leaf's overall volume.

### 2.2. Subcellular and sub-organelle metrics at the cellular level

We collected or calculated the average fractional occupancy, volume and number of organelles per cell (plastids, cytosol, mitochondrion, ER, Golgi, peroxisome and vacuole) for each of the cell types present in reference leaf 6 (see Supplementary Table S1, sheet S1.1 and a summary in Supplementary Table S2).

As shown in Supplementary Table S2, we lack data on the metrics of subcellular organelles in vein cells, bundle sheath cells and some epidermal cells (guard cells and trichome) except for plastid volumes and plastid numbers per cell, for which more abundant data are available. Supplementary Table S1, sheet S1.1 includes information on data availability, literature references as well as comments. More detailed information at the sub-organelle level can be found in Supplementary Table S1 for mesophyll chloroplasts (envelope, stroma, thylakoid, etc.; lines #337 to #373) and for mitochondria (intermembrane space and matrix space; lines #696 to #728).

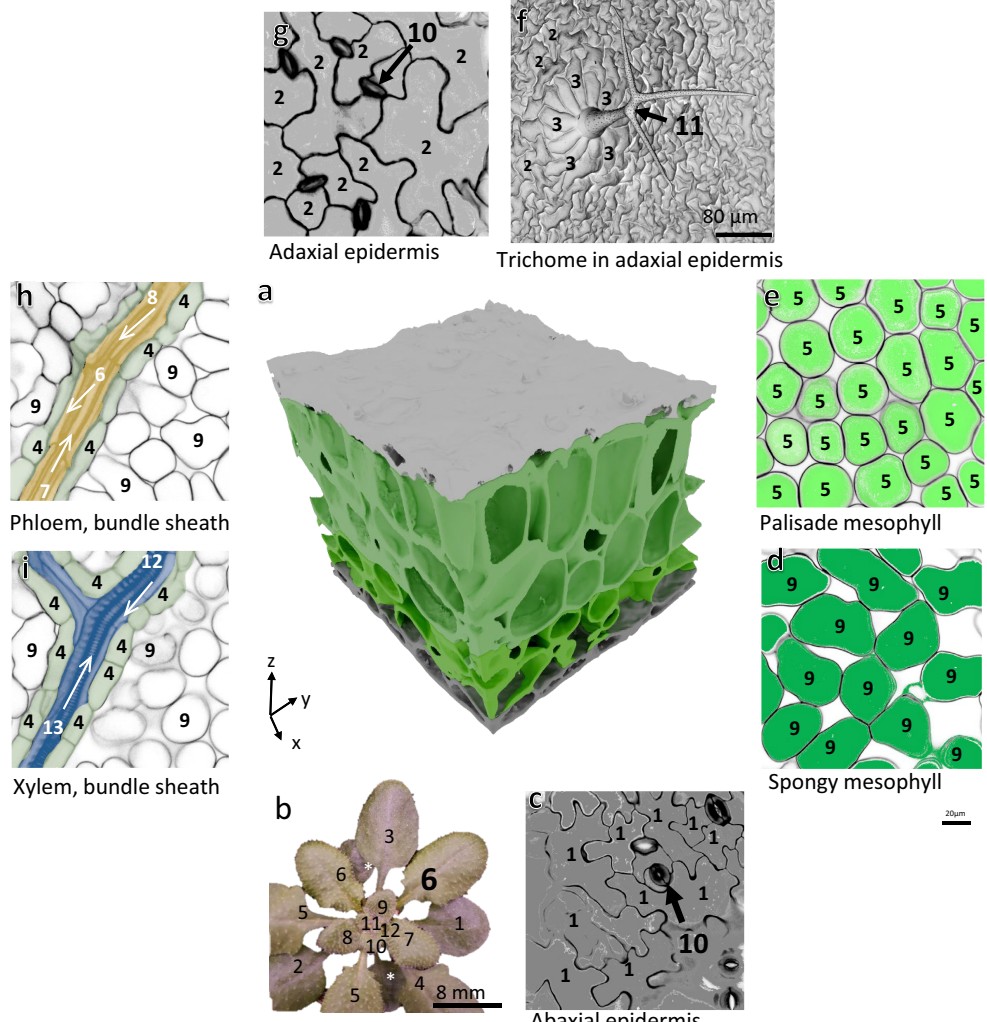

**Figure 1.** Arabidopsis leaf 6 anatomy at 21 days after initiation. (a) 3D rendering of reconstructed leaf 6 (21 days after initiation) based on multiphoton laser scanning microscopy data (provided by Nathalie Wuyts). The 3D reconstruction corresponds to a leaf piece located approximately midway between the leaf midvein and margin. (b) Arabidopsis rosette: numbers indicate the true leaf numbers in their order of formation; asterisks indicate cotyledons. See Section 4 for experimental growth conditions. (c–i) Multiphoton laser scanning microscopy images showing the different cell types in the Arabidopsis leaf in the (*x*,*y*) plane, as indicated in the pictures (adaptation from original data provided by N. Wuyts). (c,g) Abaxial (bottom) and adaxial (top) epidermis, respectively, with epidermal pavement cells and stomata complexes. (f) SEM image showing a trichome cell and its basal cells (image courtesy of Michèle Crèvecoeur). (h,i) Longitudinal section of a minor vein with phloem and xylem, respectively. Numbers in (c–i) refer to cell types (in alphabetical order): 1, abaxial (bottom) epidermal pavement cells; 2, adaxial (top) epidermal pavement cells; 3, basal trichome cells; 4, bundle sheath cells; 5, palisade mesophyll cells; 6, phloem companion cells; 7, phloem parenchyma (transfer) cells; 8, phloem sieve elements; 9, spongy mesophyll cells; 10, stomata cells; 11, trichome cells; 12, xylem parenchyma cells; 13, xylem tracheids. Cambial vein cells and hydathode cells are not displayed here.

## 2.3. Subcellular and sub-organelle metrics at the leaf level

To calculate the volume occupied by the different cellular organelles at the leaf level, we considered only cells for which complete subcellular data were available (See the summary in Supplementary Table S2). We thus simplified the leaf, representing it as an assemblage of epidermal pavement cells and mesophyll cells. This simplification is relevant as these cells make up 94.4% of the total cellular volume of the leaf. For these cells, the vacuole volume was used as a free variable to adjust the final cell volume to 100%. The results from Supplementary Table S1, sheets S1.8 and S1.9 are summarised in Table 3., with data expressed per gram fresh weight and relative to total chlorophyll content. These data are also graphically represented in Figure 2c (https://public.flourish.studio/visualisation/13318228/). Below, we consider additional calculations for the different organelles.

*Note*: Data correspond to the cell types for which fractional volume occupancy could be calculated for all subcellular organelles

(i.e., epidermal pavement cells and mesophyll cells, corresponding to 94.4% of the cellular volume in the leaf). Data for ER and Golgi are provisional (see the text and Supplementary Table S1, sheet S1.4). Volume per g LFW was calculated using the number of leaf 6 per g LFW (see Supplementary Table S1, sheet S1.1, line #17); volume per mg Chlorophyll was obtained using the conversion factor 1.2 mg chl/g LFW (see Supplementary Table S1, sheet S1.1, line #926).

### 2.3.1. Chloroplast/plastid.

According to the data (Supplementary Table S1, sheet S1.1, lines #199 to #333 and summarised in Supplementary Table S1, sheet S1.8), two thirds (63%) of the chloroplast volume is provided by the palisade cells, and the remaining third (30%) by spongy cells. This is because the number of palisade cells is higher than the number of spongy cells, not because palisade cells contain more chloroplasts than spongy cells as sometimes erroneously reported in textbooks. Bundle sheath cell chloroplasts

**Table 1.** Arabidopsis reference leaf 6 metrics.

| | value | Exp. error | Units |
|---|---|---|---|
| *Arabidopsis thaliana* col-0 leaf n°6 area 21 DAI | 121 | 17.8 | mm$^2$ |
| Specific leaf area per unit leaf dry weight | 41 | 11 | mm$^2$/mg LDW |
| Specific leaf volume excluding air space per unit leaf dry weight | 8.1 | 1.1 | mm$^3$/mg LDW |
| Specific cell number (excluding vein cells) | 145 | 38 | number/µg LDW |
| Leaf n°6 fresh weight (LFW) | **22** | | mg |
| Leaf n°6 dry weight (LDW) | **3** | | mg |
| Leaf n°6 volume (121 mm$^2$) excluding air space | **24** | | µL |
| LFW/LDW | **7.45** | | ratio |
| Leaf area per unit leaf fresh weight | **5,500** | | mm$^2$/g LFW |
| Leaf fresh weight per unit leaf area (g/m$^2$) | **182** | | g LFW/m$^2$ |
| Leaf fresh weight per unit leaf area (mg/cm$^2$) | **18.2** | | mg LFW/cm$^2$ |
| Number of fully expanded leaves n°6 per gram fresh weight | **45.5** | | number of leaves 6 per g LFW |
| Mass of chlorophyll a + b per unit leaf fresh weight | 1.2 | 0.1 | mg/g LFW |

Note: Reference macroscopic metrics for Arabidopsis thaliana Col-0 leaf 6, 21 days after growth initiation, stage 1.09 according to Boyes et al. (2001) for plants grown under reference conditions (16 h light period, 166 µmol photons m$^{-2}$ s$^{-1}$, well-watered plants, 22°C, relative humidity 72%) (Wuyts et al.,2012). Figures in bold are the results of calculations; see Supplementary Table S1, sheet S1.1 and Section4 for details and references.

**Table 2.** Arabidopsis thaliana leaf 6 cellular metrics under reference conditions.

| Cell name (alphabetical order) | Average vol. (pL) | Number in leaf 6 | Total volume in leaf 6 (µL) | Volume (µL/g LFW) | Volume (µL/mg chlorophyll) | % (cell number) | % (volume) |
|---|---|---|---|---|---|---|---|
| Abaxial (bottom) epidermal pavement cells | 19 | 58,500 | 1.1 | 49.6 | 42.6 | 7.7 | 4.5 |
| Adaxial (top) epidermal pavement cells | 39 | 43,000 | 1.7 | 75.0 | 64.5 | 5.6 | 6.9 |
| Basal trichome cells | 20 | 1,400 | 0.03 | 1.3 | 1.1 | 0.2 | 0.12 |
| Bundle sheath cells (second + higher-order veins) | 10 | 63,000 | 0.64 | 28.7 | 24.7 | 8.2 | 2.6 |
| Cambial vein cells | n.a. | n.a. | n.a. | n.a. | n.a. | n.a. | n.a. |
| Hydathode cells | n.a. | n.a. | n.a. | n.a. | n.a. | n.a. | n.a. |
| Palisade mesophyll cells | 100 | 157,500 | 16 | 716 | 616 | 21 | 65 |
| Phloem companion cells | 0.85 | 49,000 | 0.04 | 1.9 | 1.6 | 6.4 | 0.2 |
| Phloem parenchyma cells | 0.63 | 78,000 | 0.05 | 2.2 | 1.9 | 10 | 0.2 |
| Phloem sieve elements | 0.14 | 23,000 | 0.0032 | 0.15 | 0.13 | 3 | 0.01 |
| Spongy mesophyll cells | 60 | 75,000 | 4.5 | 204 | 176 | 9.8 | 18.4 |
| Stomata guard cells | 0.8 | 80,000 | 0.06 | 2.9 | 2.5 | 10.5 | 0.26 |
| Trichome cells | 3,800 | 120 | 0.46 | 20.7 | 17.8 | 0.02 | 1.9 |
| Xylem parenchyma cells | 0.5 | 104,000 | 0.06 | 2.5 | 2.2 | 13.7 | 0.23 |
| Xylem tracheids (lumen) | 0.4 | 32,000 | 0.01 | 0.5 | 0.5 | 4.2 | 0.05 |
| | | | | | | 100 | 100 |
| Total epidermal cells | | 183,000 | 3.3 | 150 | 130 | 24 | 14 |
| Total photosynthetic cells∗ | | 295,000 | 21 | 950 | 820 | 39 | 86 |
| Total vein cells (P + X†) | | 286,000 | 0.2 | 7.4 | 6.3 | 37 | 0.7 |
| Total for all leaf | | 764,000 | 24 | 1,100 | 950 | 100 | 100 |

Note: This is a summary of Supplementary Table S1, sheets S1.1 and S1.2. See Supplementary Table S1, sheet S1.7 for details of calculations. Values resulting from calculations were rounded. A leaf fresh weight of 1 g in the reference conditions described by Wuyts et al. (2012) corresponds to 45.5 individual reference leaf 6 (leaf surface 121 mm$^2$), and to 1.2 mg of chlorophyll (see Table1).
∗Bundle sheath cells + palisade mesophyll cells + spongy mesophyll cells.
†P + X: phloem + xylem parenchyma cells + tracheids.

(the only organelle for which data are available for this cell type) represent only 6% of the total leaf chloroplast volume because they contain a low number of chloroplasts per cell and the number of bundle sheath cells is low compared to mesophyll cells. The combined volume of chloroplasts from all photosynthetic cells corresponds to 10% of the total cellular volume of the leaf. As expected, chloroplasts from epidermal cells contribute only marginally (less than 1%) to this total leaf chloroplast volume. Sub-organelle data (for thylakoid, stroma, starch, lumen, etc.) are also provided in Supplementary Table S1, sheet S1.1 (e.g., thylakoid volume per g LFW; line #348).

**2.3.2. ER and Golgi.** The volumetric data presented for ER and Golgi remain provisional as few studies were available, and the

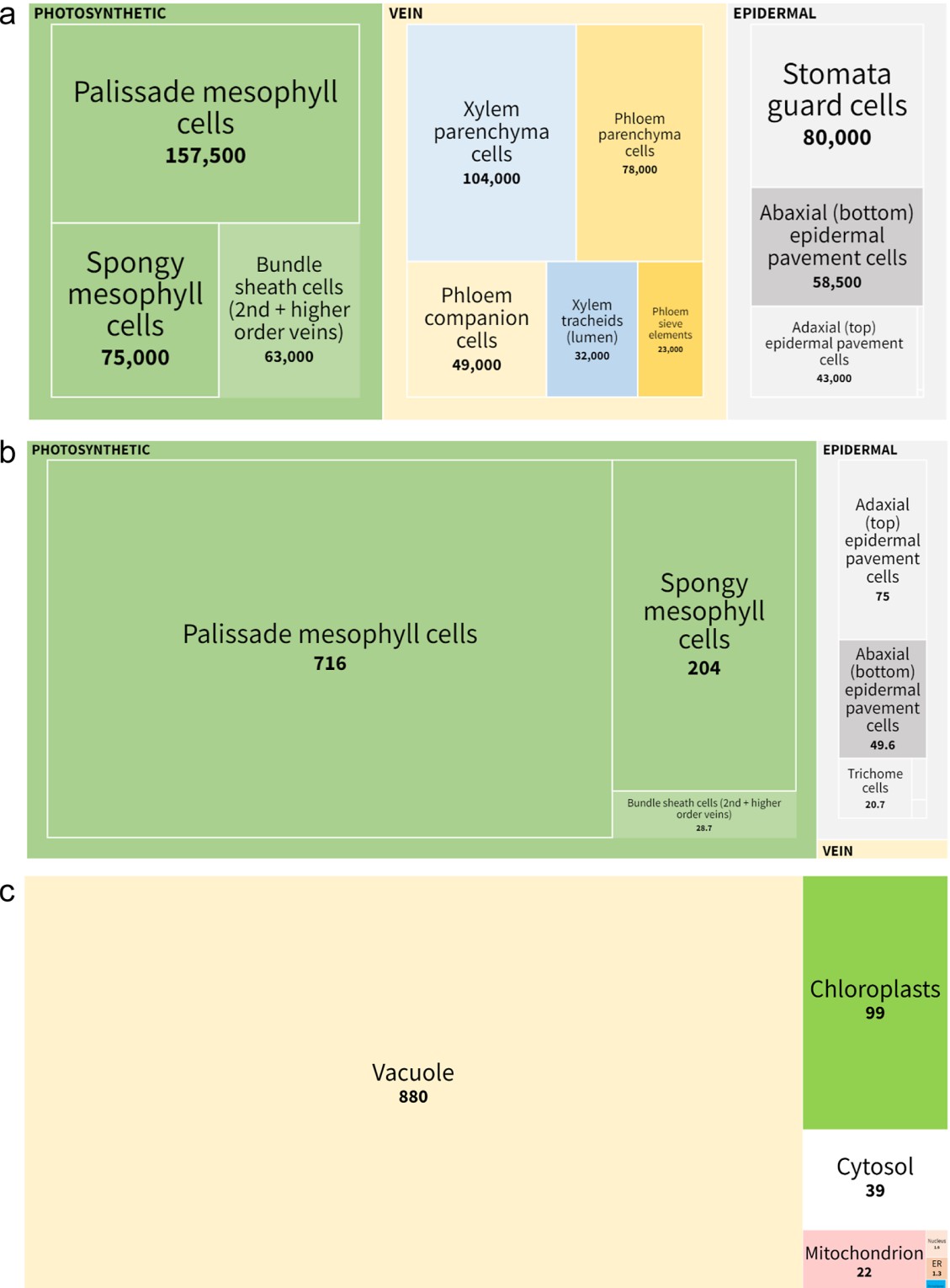

**Figure 2.** Graphical representation of the cellular and subcellular metrics of leaf 6. (a) Cell numbers; (b) volume occupancy in Arabidopsis reference leaf 6 in μL/g LFW. Growth conditions are 21 DAI, 16H light, 166 μmol photons m⁻² s⁻¹, 20.5°C, relative humidity 72%. See Table 2 for further details. (c) Relative subcellular volumes in an Arabidopsis reference leaf 6 in μL/g LFW. As data were incomplete for the other cell types (see Supplementary Table S2), the leaf was simplified as an assemblage of mesophyll cells and epidermal pavement cells (representing 94% of the leaf volume; see Table 3.). Figure generated using Flourish (https://flourish.studio/).

information presented was derived from partial analysis of an epidermal cell. For further details, see Supplementary Table S1, sheet S1.4 and the references therein, Supplementary Table S1, sheet S1.1 (lines #449 to #460) and Section 4.

**2.3.3. Mitochondria.** The mitochondria volume of the leaf is split between mesophyll cells (89%) and epidermal cells (11%) (Supplementary Table S1, sheet S1.8, lines #63 to #76). For many cell types (stomata guard cells, trichome cells and vein cells), the contribution

**Table 3.** Subcellular volumetric data for *Arabidopsis thaliana* reference leaf 6.

| Subcellular compartment volume in total epidermal pavement cells and mesophyll cells from leaf 6 | µL in one leaf 6 | µL/g LFW | µL/mg Chl | % |
|---|---|---|---|---|
| Chloroplasts | 2.2 | 99 | 85 | 9.4 |
| Cytosol | 0.9 | 39 | 34 | 3.8 |
| ER | 0.03 | 1.3 | 1.1 | 0.1 |
| Golgi | 0.002 | 0.09 | 0.08 | 0.009 |
| Mitochondrion | 0.5 | 22 | 19 | 2.1 |
| Nucleus | 0.03 | 1.6 | 1.3 | 0.15 |
| Peroxisome | 0.02 | 0.9 | 0.8 | 0.09 |
| Vacuole | 19 | 880 | 760 | 84.3 |
| Total cellular volume in epidermal pavement cells and mesophyll cells from leaf 6 | 23 | 1,050 | 900 | 100 |

to the leaf's mitochondrial pool could not be calculated due to missing data. The mitochondrial intermembrane space and matrix volumetric data presented here were derived from Fuchs et al. (2020). The detailed calculations can be found in Supplementary Table S1, sheet S1.1, lines #698 to #711. It should be noted that some of these values correspond to estimations based on data from animal cells or from heterotrophic Arabidopsis cells (see Fuchs et al., 2020 and details in the comment column in Supplementary Table S1, sheets S1.1 and S1.6).

**2.3.4. Nuclei.** The contribution of nuclei to the total cellular volume is low (0.2%), and based on cell numbers, nuclei from epidermal cells and mesophyll cells account for three fourths of the overall nuclei volume in the leaf.

**2.3.5. Peroxisomes.** Peroxisomes are important organelles involved in leaf photorespiration, lipid beta-oxidation and jasmonate synthesis. Nevertheless, they represent a very small volumetric fraction of the leaf (less than 0.1%; Supplementary Table S1, sheet S1.8, line #106 and Table 3.). Interestingly, peroxisomes appear to be as numerous as chloroplasts in mesophyll cells (see Supplementary Table S1, sheet S1.1, lines #798 and #801), this like-for-like matching reflects the strong functional interplay between these two organelles during photorespiration in C3 plants.

**2.3.6. Vacuoles.** Not surprisingly, vacuoles represent the largest fraction of the cellular volume in a leaf (84%; see Table S1, sheet S1.8, lines #109 to #122 and Table 3.). This quantitative predominance has consequences when calculating metabolite concentrations (see below), depending on whether or not a metabolite is present in the vacuole, is concentrated in this compartment or an equilibrium is maintained with the cytosol. Data on vacuole volume lack information related to vein cells and bundle sheath cells. As these cells represent a very small fractional volume of the leaf (0.7%), they do not contribute much in quantitative terms to the total vacuole volume within the leaf.

## 2.4. Calculation of membrane surfaces and cell wall volume

As membranes support important biological processes (e.g., metabolites exchanges, hormone transport or lipid metabolism), the membrane surface area is an important metric to consider in functional analyses. Thanks to the data presented above and additional parameters extracted from the literature (see Supplementary Table S1, sheet S1.5), the surface areas of the different cell types and organelles could be calculated by modelling base on simple geometric shapes (spheres, capsules, jigsaw puzzle

pieces, tors and cylinders). Surface area values were extrapolated to produce a figure for the entire leaf using cell numbers per leaf (see above).

It was possible to calculate the relative contributions of the plasma membrane and the different organelle membranes to the overall membrane content of the leaf. According to the data collected and additional calculations, the total surface area of thylakoid membranes represents 800-fold the leaf's surface area (Supplementary Table S1, sheet S1.5, line #274): this value is within the upper limit of the reported range for spinach ('300 m$^2$ leaf m$^{-2}$ (other estimates 835)'; see page 56 in Lawlor, 2001). We also derived other ratios. For example, plasma membrane and mitochondrion outer membrane surface area represent 30-fold and 12.4-fold the leaf area, respectively (see Supplementary Table S1, sheet S1.5). The ratio of internal chloroplast membranes (thylakoids) to chloroplast outer membrane is 28, close to the ratio of chloroplast envelope membrane to leaf area (28.8). The total surface area of chloroplasts in reference leaf 6 (35 cm$^2$) is close to the total plasma membrane surface area for all the cells from leaf 6 (33 cm$^2$). Whether these ratios represent optimisation principles in plants or are specific to Arabidopsis in the reference conditions chosen remains to be determined. If such constraints do exist, they might prove useful in modelling plant development and metabolism.

Cell wall volume was also calculated for each cell type, assuming a thickness of 1 µm for xylem tracheary elements (Wei et al., 2022), 0.3 µm for sieve elements (Froelich et al., 2011) and 0.15 µm for other cells (see Supplementary Figure S2 and Supplementary Table S1, sheet S1.5, lines #4 to #19). We used 'microscopic' data (cell areas and cell numbers) to estimate the relative contribution of the overall cell wall volume to the total cellular volume of the leaf (2.7%; see Supplementary Table S1, sheet S1.5, line #179). This value is very close to the 2.9% determined from macroscopic data (see Supplementary Table S1, sheet S1.6, line #128; Borniego et al., 2020), lending strong support to the calculation method, despite the assumptions made to simplify cell shapes.

## 2.5. Conversion factors and metabolite concentrations (in µmol L$^{-1}$)

In this part, we present examples of use of the volumetric data provided in Supplementary Table S1 to estimate absolute metabolite concentrations, and how these concentrations vary depending on conditions and across experimental studies. For this purpose, we derived conversion factors to transform data provided in their original units to a standard unit. This transformation is an essential step for use of data with constraint-based metabolic models.

Calculating the conversion factors is complex, requiring several types of independent information: (i) the volume of the different cellular and subcellular compartments, (ii) the original units used, (iii) the original biological material (e.g., whole leaf and isolated chloroplast), (iv) the biological 'fraction' used for the quantification (e.g., 'whole leaf' or 'thylakoids'), (v) the method used (e.g., non-aqueous fractionation), (vi) hypotheses on the cellular and subcellular localisation(s) of the metabolites and (vii) relative units (e.g., mg chlorophyll/g LFW, LFW/LDW and g LFW/mm2) where LFW is leaf fresh weight and LDW is leaf dry weight.

Information types (i)–(vi) are collated in the ChloroKB database (Gloaguen et al., 2017, 2021), which has been made available online, associated with this paper as the 'Quantitative data' document (see Supplementary Table S3; this document is also available from the ChloroKB website, download page, item 4, http://chlorokb.fr). This file contains standard molecule names, molecular weights, cross references and identifiers in chemical databases (Kegg, Chebi and Pubchem), as well as InChI and SMILES codes to facilitate cross-identification. Quantitative data are provided in their original units as single values or concentration ranges (minimum and maximum values measured), together with the relevant literature references, the name of the original biological material, growth conditions, analytical methods used, and a curated subcellular localisation field. All localisation information was manually inserted into ChloroKB and integrates both the nature of the biological material (e.g., isolated vacuole or whole leaf) and biological knowledge extracted from the reconstructed metabolic network of ChloroKB. We chose to neglect the contribution of epidermal and vein cells in this metabolic analysis, as they represent a relatively small proportion of the total leaf volume (0.7% for veins and 13% for epidermal cells), and little is known about their metabolism. When metabolites are distributed across several subcellular compartments, we assumed concentrations to be equal in the different compartments (i.e., the amount in nanomol/g LFW was divided by the sum of the subcellular compartment volume in $\mu$L/g LFW, to obtain an average concentration in nanomol/$\mu$L, which was then converted into $\mu$M). This assumption will be incorrect for some metabolites, such as those that concentrate in some compartments. However, in most cases, we do not have access to this level of information. Our calculations, therefore, provide an order of magnitude of the concentrations of such metabolites, and modellers can set a reasonable range for variation around this value.

To convert crude data exported from ChloroKB (Supplementary Table S3) into concentrations (in $\mu$M), we developed a Python script (see Supplementary Method 1) drawing information from several tables derived from Supplementary Table S1, sheet S1.1. These tables are summarised in four sheets in Supplementary Table S4. Concentrations expressed as $\mu$M are listed in the last four columns of Supplementary Table S5. Users who prefer values in nanomol/g LFW can find them in columns Y, Z and AA. Unicity ($\mu$M) is useful when seeking to compare metabolite abundances across studies, subcellular compartments and conditions (see Supplementary Figure S3 graphs for a selection of central metabolites). The data (each bar in Supplementary Figure S3 corresponds to one measurement extracted from a publication) reveal either significant variability (e.g., 2-cis-abscisate, L-proline, sucrose and trehalose 6-phosphate) or relative stability across studies/conditions (e.g., L-aspartate, L-alanine and UDP-D-glucose). Thanks to the different molecule identifiers indicated, this file can be used to constrain models by applying realistic boundaries for metabolite concentrations – for example, for Thermodynamic Flux Balance Analysis (Henry et al., 2007) – or to feed Resource-Balanced Analysis

models (Goelzer & Fromion, 2017). Model predictions can also be compared to these 'real' experimental data. Finally, the data may help to define the physiological operating points for complex enzyme kinetics and to determine the physiologically relevant concentration range for substrates, products or effectors when testing enzymatic activity *in vitro* (Curien et al., 2009; van Eunen et al., 2010). These data contribute significantly to simplifying calculations of protein/substrate and protein/protein stoichiometry.

## 3. Discussion

The main goal of this article was to collect, curate and organise quantitative data related to Arabidopsis leaves to provide the plant community with an aggregated, harmonised and unified dataset, as part of efforts to build a quantitative plant leaf atlas. There are, thus, three important facets to this work: first, a large body of references and state-of-the-art quantitative data are made immediately accessible for further use; second, calculations were made on a large ensemble of data derived from the source information (the results are not available elsewhere); and third, mathematical formulas embedded in Supplementary Table S1 can be used to treat new data as they become available. Thus, any modification, following the acquisition of new data, will be propagated throughout the document and lead to modified output values.

Leaf 6 was used as a reference here because it is the first adult leaf to develop on the Arabidopsis plantlet. However, leaf area is proportional to LFW (Massonnet et al., 2010) and leaf vein length is proportional to leaf area (Dhondt et al., 2012). Consequently, our results are certainly valid for the other leaves (7, 8, 9 etc.) as long as the rosette leaves do not enter into senescence. Values for leaf 6 can be easily modified to reflect other leaves simply by changing the leaf area listed in Supplementary Table S1, line #6.

In some instances, we collated data from experiments performed in non-standard experimental conditions – at different light intensities, or with a different light period during growth. Despite this, the values collected could be used in calculations to provide first estimates of metrics such as cell or organelle numbers, or areas. More work will be required to provide a comprehensive dataset for the various growth conditions used in experimental studies, and thus to quantitatively appreciate the leaf's plasticity and its limits at the cellular and subcellular levels.

The present compendium provides a framework for various uses such as in single-cell analyses, quantitative modelling and biochemical studies – where quantitative data are powerful aids to test hypotheses. The information contained in the Supplementary Tables can be used to perform additional calculations or to build quantitative atlases for other tissues in Arabidopsis or other plants. The data presented in Supplementary Table S1 will be updated regularly along with ChloroKB knowledge base content (*ca.* twice a year) and made available through the ChloroKB website.

Over the course of this work, we observed that lack of a single type of data, such as a fresh weight/dry weight ratio or absolute quantification (of the 100% value), for example, made published data impossible to use. We thus recommend that authors provide as many relative units as they can in their articles (LFW/LDW; leaf surface/g LFW; mg Chlorophyll per g LFW; etc.) so that quantitative data can be extracted to allow comparisons between studies. We also encourage researchers to provide an accurate description of their experimental conditions – as specified in Poorter et al. (2012) – in the machine-readable format described by Hannemann et al. (2009). Adopting these habits will greatly facilitate automated data mining.

The present initiative to integrate heterogeneous data accumulated over a number of years across many laboratories is an illustration of the tension that exists between modelling and hypothesis-driven experiments. Synergism in the future would help alleviate this tension. The Open Science initiative is expected to tackle this issue, thanks to data Fairification – that is, making data Findable Accessible, Interoperable and Reusable (Barton et al., 2022; Saint Cast et al., 2022). We consider our work to be a contribution in this direction. Experimental scientists undoubtedly have vast amounts of unpublished quantitative data that were not deemed necessary to support their discoveries. These data represent a treasure trove. Biologists using hypothesis-driven approaches can also gather information on what modellers need to know and how to format data for publication to facilitate future use. Editors should be encouraged to welcome quantitative data as supplementary information, even if the data are not directly required to support a given concept or new findings. Descriptions remain an important intrinsic part of the science of biology, and quantitative descriptions are more demanding in terms of standards and references. We hope this work will stimulate experimental investigators to collect comprehensive volumetric/area data, regardless of their overarching scientific question. This key information will help us to better understand the capacities of plant metabolic networks as a whole in relation to plant development. It is especially important to consider the bigger picture given our current capacity to massively produce data using sophisticated modern tools such as plant phenotyping platforms, imaging techniques (Midorikawa et al., 2022; Oi et al., 2017; Pipitone et al., 2021; Weiner et al., 2022), segmentation tools (Harline & Roeder, 2023; Wolny et al., 2020), proteomics (Mergner et al., 2020), laser microdissection (Balasubramanian et al., 2021) and single-cell analyses (de Souza et al., 2020; Seyfferth et al., 2021).

Through the efforts presented here, the framework for a detailed quantitative description of a reference Arabidopsis leaf at both the cellular and subcellular levels is set in place. Extracting data from the literature is time-consuming, and the presentation made here aims to facilitate this step for a range of research purposes, such as modelling, biochemical analyses and/or any work requiring absolute quantification. The present data compendium paves the way towards a more comprehensive quantitative atlas of whole Arabidopsis plants, and offers a new repository of quantitative data that are valuable per se as constraints when investigating plant metabolism. We hope that this shared resource will pique the interest of the plant modelling community.

## 4. Methods

*Data selection*: The procedure used to select the most relevant data is illustrated in Supplementary Figure S1.

*Data organisation*: The data are presented in a single Excel file (Supplementary Table S1) comprising 11 sheets named as follows: Sheet S1.1 (complete data); Sheet S1.2 (veins data); Sheet S1.3 (minor veins cell numbers) dedicated to cells in minor veins, giving number per cross-section; Sheet S1.4, dedicated to calculations of ER and Golgi subcellular volumes; Sheet S1.5 (areas and cell wall) for calculation of areas of cells and cell wall volumes; Sheet S1.6 (additional data); Sheet S1.7 (summary of cell numbers and volumes); Sheet S1.8 (organelle volumes); Sheet S1.9 (summary of organelle volumes in a single leaf); Sheet S1.10 (Legend to table) and Sheet S1.11 (abbreviations). Data in the different tables are linked across sheets to provide the final computation in Supplementary Table S1, sheets S1.1 and S1.5, and in the summary tables.

The summary tables consist in a selection of data from Supplementary Table S1.

Supplementary Table S1, sheet 1.1 is organised as follows: 1, macroscopic data (lines #5 to #17), with leaf surface area, LFW/LDW ratio etc.; 2, cellular metrics (lines #19 to #101); 3, sap and apoplast data (lines #103 to #108); 4, subcellular metrics (lines #200 to #922); 5, data on chlorophyll for conversions (lines #924 to #938); 6, conversion factors (lines #941 to #1004). For microscopic data for the different cells, cell types were sorted alphabetically, and a colour code was applied (detailed in the Legend sheet). The organisation is as follows: 1, cell densities (number per mm$^2$; lines #21 to #37); 2, cell numbers (for each cell type) in a single reference leaf 6 (lines #39 to #62); 3, single-cell volumes (lines #64 to #80); 4, total cell volume per leaf for each cell type (lines #83 to #101); 5, sap volume (lines #103 and #104); 6, apoplast volume (lines #106 to #18); 7, air space volume (lines #112 to 114); 8, cell volumes per m$^2$ leaf surface area (lines #120 to #136); 9, cell volumes per g LFW (lines #138 to #160).

For subcellular compartments, data were organised alphabetically, applying the colour code detailed in the Legend sheet in Supplementary Table S1: 1, chloroplast or plastid (lines #199 to #333); 2, plastidial sub-compartments (stroma – lines #337 to #340; thylakoid – lines #342 to #348), thylakoid lumen (lines #350 to #353), plastoglobuli (lines #355 to #358), envelope (lines #360 to #363), nucleoid (lines #365 to #368) and starch (lines #370 to #373). The data relate only to mesophyll cell chloroplasts, as no data were available for plastids in other cell types; 3, cytosol (lines #379 to #445); 4, ER (lines #447 to #506); 5, lipid droplets (lines #509 to #513); 6, Golgi (lines #515 to #573); 7, mitochondria (lines #576 to #728); 8, nucleus (lines #731 to #788); 9, peroxisome (lines #790 to #860); 10, vacuole (lines #863 to #922). For each subcellular compartment, the data for individual cell types are indicated when available.

### 4.1. Leaf parameters

*Reference leaf*: We used the three-dimensional images of Arabidopsis leaf 6 obtained by Wuyts et al. (2010) and Wuyts et al. (2012) using multiphoton laser scanning microscopy (see Figure 1). Wuyts et al.'s (2012) study provides the most extensive anatomical and volumetric description of mesophyll and epidermal tissues in Arabidopsis leaf. In their study, Arabidopsis plants were grown in a controlled environment under well-watered conditions and 'high cumulative light' (9.6 mol m$^{-2}$ day$^{-1}$, 16 h light, i.e., 166 µmol photons m$^{-2}$ s$^{-1}$, 72% relative humidity). The leaf was cut after 1 hour in the light period. The data were collected for the nearly fully expanded leaf 6 (121 mm$^2$), 21 days after initiation (Arabidopsis-Boyes-stage 1.09; Boyes et al., 2001). This corresponds to leaf developmental stage (LDS) 0.84 (Rowan & Bendich, 2009). LDS = S/MS + D, where S is leaf area at the time used for measurements, MS is the maximum surface area (here 144 mm$^2$) and D is the number of days after the leaf has reached full expansion. Here, D = 0 because the leaf used for multiphoton laser scanning microscopy had not reached full expansion in Wuyts et al. (2012), with a leaf surface area of 121 mm$^2$.

From several parameters explicitly provided in Wuyts et al. (2012), it was possible to calculate others, such as the volume of leaf 6 (line #12 in Supplementary Table S1.1). LFW could not be directly derived, but was estimated from another article from the same research group (Massonnet et al., 2010), presenting the relation between rosette surface area and fresh weight. An issue was encountered for conversion of the data into per mg chlorophyll

as Wuyts et al. (2012) did not quantify chlorophyll. This problem was circumvented thanks to the availability of a thorough analysis of chlorophyll content and how it changes over the course of the Arabidopsis life cycle (Nath et al., 2013). The experiments in Nath et al. (2013) were performed under conditions close to those in Wuyts et al. (2012), with long days (16 h). However, they were not strictly identical; for example, light intensity was 100 μmol photons $m^{-2}$ $s^{-1}$ in Nath et al. (2013) compared to 166 μmol $m^{-2}$ $s^{-1}$ in Wuyts et al. (2012). This may have introduced a bias in our calculations as chlorophyll content is known to increase with light intensity. However, the change is not large and is close to experimental error (from 1 to 1.2 mg/g LFW with light increasing from 100 to 1,000 μmol photons $m^{-2}$ $s^{-1}$; see Figure 8 in Zhang et al., 2016). The value of 1.16 mg Chlorophyll/g LFW for plants at the same developmental stage, therefore, appears to be a reasonable estimate (see chlorophyll data in Supplementary Table S1, sheet S.1, lines #924 to #938).

### 4.2. Cell types, numbers and volumes

#### 4.2.1. Cell numbers in a single reference leaf 6.
Plant leaves contain 15 distinct cell types, including specialised cells such as hydathode cells, phloem sieve elements (anucleate cells) and dead cells (xylem tracheids) (Table 2). Cell-type densities relative to leaf surface area for epidermal cells (pavement cells and stomata cells) and mesophyll cells from Wuyts et al. (2012) were used to calculate the number of these four cell types in reference leaf 6. Trichome cell numbers were derived from in-house observations. The assumption that there were 12 basal cells per trichome cell was applied (Ebert et al., 2010). The density of bundle sheath cells (i.e., the cells surrounding minor veins) was more complex to evaluate as their number varies depending on vein order, with the leaf's mid-rib being devoid of bundle sheath cells. Numbers of these cells were therefore determined based on vein length, cell length and number of cells per vein cross-section. Although bundle sheath cells are not vein cells, we relied on vein metrics to calculate their number (see details in Supplementary Table S1.2). The results (cell numbers and volumes) were included in Supplementary Table S1.1.

The true vein cell number in Arabidopsis leaf 6 (i.e., excluding bundle sheath cells) or relative to leaf surface area was derived from several independent studies. Vein length per leaf area is provided in Caringella et al. (2015). As vein length is proportional to leaf area (Dhondt et al., 2012), vein length could be computed for Arabidopsis reference leaf 6. The number of individual vein cell types per minor vein cross-section is well documented. However, the final number varies depending on the study. To account for this variability, averaged values were used here, as computed in Supplementary Table S1.3. The total number of each vein cell type in reference leaf 6 was estimated by multiplying their number expressed per cross-section (Supplementary Table S1.2, lines #33 to #74) by the length of the veins (lines #9 to #19) and then dividing by the lengths of each cell type. The latter values were obtained from observations of longitudinal sections of living material (see the references in Supplementary Table S1.2, lines #25 to #31), and the same length was assumed for the different vein orders. Data published for primary and secondary veins are scarce; as a result, single pictures were used to calculate cell numbers for each cross-section. The mid-rib was modelled as a cone: the number of cells was calculated for a cylinder, and as a first approximation, the result was divided by 3 (the volume of a cone is one third that of the cylinder in which it is included).

#### 4.2.2. Cell volumes in reference leaf 6.
Average cell volumes are available for epidermal pavement cells and mesophyll cells in Wuyts et al. (2010) and Wuyts et al. (2012). Guard cell volume was taken from Chen et al. (2012), and trichome cell volume from Gutierrez-Alcala et al. (2000). Volumes for other cells (vein cells and bundle sheath cells) were estimated from longitudinal cuts and from measurements of cell cross-sections (see details in the 'comment' and 'method' columns in Supplementary Tables S1.2 and S1.3), assuming these cells to be cylindrical. The averaged cross-section was measured using ImageJ. The same cell sizes were assumed to apply to the different vein orders.

By multiplying the volume of each cell type by the number of cells of this type in leaf 6, the volume contribution of each cell type in leaf 6 was obtained.

#### 4.2.3. Apoplast fluid volume per g leaf fresh weight.
The value published by Borniego et al. (2020) (21.4 μL/g LFW) was used (Supplementary Table S1.1, line #162).

#### 4.2.4. Volume of cells per g LFW.
Leaf 6 fresh weight (22.3 ± 4.5 mg; see line #10 in Supplementary Table S1.1) was derived from data in Wuyts et al. (2012) and Massonnet et al. (2010) with a fresh weight/dry weight ratio of 7.97 (see the 'comment' column in Supplementary Table S1.1, line #10). Volumes of cells per reference leaf 6 (in μL/leaf 6) were converted into volumes per g LFW by multiplying the cell volume per leaf 6 by the number of leaves per g LFW (45.5 leaves/g LFW; see line #17 in Supplementary Table S1.1).

### 4.3. Subcellular compartment volumes

The following compartments were considered (see the Legend sheet in Supplementary Table S1): chloroplast (or plastid) envelope membranes, chloroplast stroma, thylakoid membranes, thylakoid lumen, cytosol, endoplasmic reticulum, Golgi, mitochondrion intermembrane space, mitochondrial matrix, peroxisome and vacuole.

3D reconstructions of the subcellular organisation inside the different Arabidopsis mature leaf cells are not yet available (partial data are available in Pipitone et al., 2021) for cotyledon chloroplasts, and (Midorikawa et al., 2022) for young mesophyll cells in Arabidopsis plantlets (grown on 1% sucrose). Data thus had to be collected from several independent studies. The references, methods used and growth conditions are indicated in Supplementary Table S1.1. Vacuole volume for mesophyll cells and epidermal pavement cells was used to adjust the final volume – that is, it was taken as the cellular volume remaining after the volumes of all the other compartments had been deducted from the total cell volume.

### 4.4. Chloroplasts

#### 4.4.1. Epidermal pavement cell chloroplasts.
Epidermal pavement cells contain about 10 chloroplasts per cell (9–15; Barton et al., 2016, 2018). Based on images of epidermal chloroplasts published by this group, an ellipsoid shape with radii of 2, 2 and 0.5 μm was assumed, corresponding to a volume of 8.4 fL. The abaxial (lower) and adaxial (upper) epidermis were assumed to contain the same number of chloroplasts. Fractional volume occupancy for chloroplasts was thus 0.45% for abaxial epidermal cells and 0.22% for adaxial cells as adaxial cells are larger than abaxial cells.

#### 4.4.2. Bundle sheath cell chloroplasts.
Chloroplast parameters for bundle sheath cells could be estimated from Kinsman and Pyke (1998), with 20 chloroplasts per cell, corresponding to a volume

of 92 fL assuming a spherical chloroplast with a radius of 2.8 μm. This volume corresponds to a fractional occupancy of 5.4% for chloroplasts in these cells.

**4.4.3. Mesophyll cell chloroplasts.** Chloroplast numbers per cell (100) and average chloroplast volume (93 fL) in mesophyll palisade cells were available from Crumpton-Taylor et al. (2012). With an average cell volume of 100 pL (Wuyts et al., 2012), the corresponding chloroplast fractional volume occupancy would be 9.3% for mesophyll palisade cells. The equivalent value is not available for spongy cells of plants grown under long-day conditions. However, palisade and spongy mesophyll cells contain the same amount of pigment per cell, and have the same rubisco activity (on a mg protein basis) (Seeni et al., 1983). It was thus assumed that they contain the same number of chloroplasts, with the same average chloroplast volume (93 fL). Based on these assumptions, the fractional volume occupancy of chloroplasts in spongy mesophyll cells is 15.5%, because spongy cells are smaller (60 pL) than palisade cells (100 pL).

**4.4.4. Stomata guard cell chloroplasts.** The volume of one guard cell was set at 0.783 pL (Chen et al., 2012) with each cell containing an average of five chloroplasts (Pyke & Leech, 1994) and (Fujiwara et al., 2018). The chloroplasts appear as 4-μm diameter spheres, total volume 33.5 fL (Fujiwara et al., 2018), with a fractional volume occupancy for the cell of 21.4%.

**4.4.5. Trichome cell chloroplasts.** Trichome cells contain leucoplasts (i.e., non-photosynthetic plastids) (Barton et al., 2018), with a size similar to epidermal cell chloroplasts (8.4 fL) (Barton et al., 2016). These organelles are numerous – 32 could be counted in the base of a trichome (see Figure 5a in Barton et al., 2018). Assuming 100 plastids for a whole trichome with a plastid volume of 8.4 fL, and an average trichome volume of 3800 pL (Gutierrez-Alcala et al., 2000), the fractional occupancy of leucoplasts would be 0.022%.

**4.4.6. Vein cell plastids.** Only partial data were available for phloem cells. Chloroplast or plastid volume in phloem can be extracted from the literature (17.3 fL for companion cell, 11.3 fL for parenchyma cell and 0.5 fL for sieve element plastids; see Cayla et al., 2015); however, the number of chloroplasts is known only for phloem companion cells (10 chloroplasts; see Cayla et al., 2015).

### 4.5. Sub-plastidial compartmentation

The fractional volumes occupied by chloroplast envelope membranes (3.9%), stroma (61%), thylakoid lumen (7.7%) and thylakoid membranes (14.9%) in mesophyll leaf chloroplasts were determined from several different measurements carried out with chloroplasts from spinach (Lawlor, 2001; Zellnig et al., 2004) and Arabidopsis (Crumpton-Taylor et al., 2012; Tolleter et al., 2017). Nucleoid volume was estimated at 1% of total chloroplast volume from Kowallik and Herrmann (1972). The starch volume was used to adjust the total volume to 100% when all the fractional volumes were added: the resulting value for starch (10.5%; Supplementary Table S1.1, line #378) only deviated slightly from the value of 15% measured in Arabidopsis chloroplasts at the end of the day (Crumpton-Taylor et al., 2012).

### 4.6. Cytosol

**4.6.1. Epidermal cell cytosol.** The fractional volume occupancy of cytosol in Arabidopsis epidermal cells has not yet been reported. Therefore, the value published for spinach epidermal cells was used (Winter et al., 1994). In that article, the compartment named 'cytosol', representing 4.96% of the cell volume, actually included the nucleus, endoplasmic reticulum and Golgi. The fractional volume occupied by the nucleus is known for Arabidopsis epidermal cells (Poulet et al., 2015) (lines #733 and #744). ER and Golgi volume were estimated to represent 3.23% and 0.323% of the cytosol volume, respectively (see ER data below). It was thus possible to calculate the volume occupied only by the cytosol in abaxial and adaxial epidermal cells (see Supplementary Tables S1.1 and S1.4 for details).

**4.6.2. Mesophyll cell cytosol.** Fractional occupancy of cytosol, including ER and Golgi, was measured in Arabidopsis mesophyll spongy cells (3.77%; Koffler et al., 2013). As for epidermal cells, it was provisionally assumed that the ER represents 3.23% and the Golgi 0.323% (see below) of the 'cytosol plus ER and Golgi' fraction. The cytosol volume fractional occupancy in mesophyll cells was then calculated after deduction of the contribution of ER and Golgi (see Supplementary Tables S1.1 and S1.4 for details). The same value was assumed for palisade mesophyll cells.

### 4.7. Endoplasmic reticulum

An approximate ER fractional volume occupancy can be derived from Bouchekhima et al. (2009), where the ER in *Nicotiana tabacum* epidermal cells was calculated to occupy 1,405 μm$^3$ of the total cytosolic volume – 41,898 μm$^3$ – that is, 3.23% of the volume of cytosol analysed. For Arabidopsis, the same occupancy of the cytosol volume for both ER and Golgi was assumed in epidermal cells and in the other cell types making up the leaf. However, this value should only be considered as an estimate, as ER fractional occupancy might depend on cell type.

### 4.8. Golgi

No quantitative data on Golgi volume were found in the literature. However, using published pictures (Boevink et al., 1998), the Golgi volume was estimated to represent one tenth of the ER volume, that is, 0.323% of the cytosol volume. The details of the corresponding calculation can be found in Supplementary Table S1.4.

### 4.9. Lipid droplets

Lipid droplets are absent from non-stressed mature leaf cells (Brocard et al., 2017), and lipid droplet fractional volume occupancy was set to zero (see lines #509 to #513).

### 4.10. Mitochondria

For mitochondrial volume and abundance, data were taken from Armstrong et al. (2006), as it provides the most complete data for four different Arabidopsis cells types (abaxial and adaxial epidermal pavement cells, and palisade and spongy mesophyll cells). The data were derived from confocal microscopy and 3D reconstructions of mitochondria following GFP-targeting of the organelle in whole plants. Based on mitochondrion volume and number in these four cell types, the fractional volume occupancy was calculated (1.37% for abaxial cells, 2.2% for adaxial pavement cells, 2.4% for palisade cells and 1.2% for spongy mesophyll cells; see Supplementary Table S1.1, lines #634 to #645). From these figures, the volume of mitochondria in each cell was determined for each of the cell types in leaf 6, and per g LFW. Data for the other leaf cell types were partial (see Supplementary Table S1.1), making it impossible to calculate their contribution to the total pool of

mitochondria in a leaf. In Armstrong et al. (2006), plants were grown under short-day conditions. Consequently, the values in Supplementary Table S1.1 might have to be re-evaluated when data become available for plants grown under long-day conditions – the reference condition chosen for data in Supplementary Table S1.

**4.10.1. Sub-mitochondrial compartmentation.** The fractional volumes occupied by the different sub-compartments of the mitochondria were calculated from data in Fuchs et al. (2020) with matrix representing 50%, membranes 34.3% and intermembrane space 15.7% of the mitochondrial volume. These data were obtained for heterotrophic cultured Arabidopsis cells; re-evaluation might be necessary when new data become available.

### 4.11. Nucleus

The volume occupied by the nucleus in the different nucleated cell types was obtained from several different studies (see the references in Supplementary Table S1.1). The fractional volume occupancy of nuclei was then easily calculated. No data were available for phloem and xylem parenchyma cells.

### 4.12. Peroxisome

**4.12.1. Mesophyll cell peroxisomes.** The fractional volume occupancy of peroxisomes in a spongy mesophyll cell (0.14%; Koffler et al., 2013) and the volume of a single peroxisome 0.97 fL (Olsen, 1998) were used to calculate the number of peroxisomes in each spongy mesophyll cell (92; see line #806 in Supplementary Table S1.1). The numbers of peroxisomes were assumed to be similar in palisade mesophyll cells when computing the fractional volume occupancy.

**4.12.2. Epidermal cell peroxisomes.** In the pictures published in Jedd and Chua (2002), each guard cell from the stomata contained at least 10 countable peroxisomes. As the number of peroxisomes in epidermal pavement cells is unknown, a provisional fractional occupancy was calculated for these cells, assuming peroxisome numbers to be the same as in stomata cells. Trichome cells contain 75–150 peroxisomes (see Figure 8a in Mathur et al., 2002). Data were not available for other cell types.

### 4.13. Vacuole

The vacuole fractional volume occupancy in the different cells was used to adjust to a final cell fraction of 1 when adding all the different fractional occupancies for the subcellular compartments (including lipid droplets – set to zero here). The values for epidermal cells (94%) and spongy mesophyll cells (79.4%) were similar to published values (94% for spinach epidermal cells in Winter et al., 1994 and 77.8% for Arabidopsis spongy mesophyll cells in Koffler et al., 2013). For other cell types except for stomata guard cells (64% of the cell volume; Tanaka et al., 2007), the vacuole volume is unknown.

## 5. Calculation of surface areas and cell wall volume

### 5.1. Calculation of surface area for abaxial and adaxial epidermal pavement cells

The surface occupied by trichome cells and their basal cells was neglected in these calculations. The fraction of leaf 6 area occupied by stomata complexes (two guard cells plus stomata pore) was first calculated on the abaxial and adaxial sides (stomata density is different on either side of the leaf; see Supplementary Table S1.5, lines #33 and #34). To calculate the periclinal (i.e., 'horizontal')

areas of abaxial and adaxial cells, the area occupied by stomata complexes was then subtracted from the total leaf area. The average periclinal area of epidermal pavement cells was obtained by dividing the remaining leaf area by the number of epidermal pavement cells (abaxial or adaxial) per reference leaf 6. Epidermal cells have complicated puzzle-like shapes. To obtain the average anticlinal (i.e., 'vertical') area of an epidermal pavement cell, the ratio between cell surface area and perimeter was first calculated. To do so, the perimeter and periclinal area of five epidermal pavement cells were measured in mature Arabidopsis leaves (Kawade & Tsukaya, 2017) using ImageJ. The average ratio between perimeter and area (14; see Supplementary Table S1.5, line #39) was used to obtain the average perimeter for the average periclinal area (see Supplementary Table S1.5, lines #40 and #50). The area of the anticlinal wall (assumed to be straight) was then determined by multiplying the average cell height (6 and 8 μm for abaxial and adaxial cells, respectively; Wuyts et al., 2012) by the average cell perimeter. The total surface area of epidermal cells was obtained by adding anticlinal and periclinal surface areas.

### 5.2. Calculation of surface area for bundle sheath cells

The bundle sheath cell was modelled as a cylinder of average length 48.7 μm (Supplementary Table S1.5, line #60) and average radius 8.2 μm (Supplementary Table S1.5, line #62). Radius was calculated from the section's surface area, assuming a circular perimeter.

### 5.3. Calculation of surface area for palisade mesophyll cells

Palisade mesophyll cells were modelled as a capsule (see Govaerts et al., 1996), which corresponds to a hollow cylinder of radius $r_c$, height $h$, plus extremities formed by two half-oblate spheroid caps of 'vertical' radius $r_s$. Parameters $h$ and $r_s$ were set to 50 and 5 μm, respectively, based on data from Wuyts et al. (2012). The radius of the cylinder ($r_c = 23.7$ μm; see Supplementary Table S1.5, line #71) was calculated using the formula for the volume of a capsule (see Govaerts et al., 1996) setting cell volume to 100 pL (Wuyts et al., 2012). The approximate area of a palisade cell was finally calculated by adding the surface area of the hollow cylinder of radius $r_c = 23.7$ μm to the area of one oblate spheroid (each extremity represents a half-oblate spheroid of vertical radius 5 μm), using Knud Thomsen's formula (see Supplementary Table S1.5, line #72).

### 5.4. Calculation of surface area for spongy mesophyll cells

Spongy mesophyll cells were modelled as spheres; radius was calculated from the volume of the cell (see Supplementary Table S1.5, line #80) and the cell's area (Supplementary Table S1.5, line #81) was derived from this radius.

### 5.5. Calculation of surface area for stomata guard cells

Stomata guard cells were modelled as symmetrical semi-tors with two discoid ends. The small radius of the tor was that of the cell (3 μm; see Supplementary Table S1.5, line #24), and the large radius (7.5 μm) was taken as the distance between the pore centre for the stomata and the cell centre. Small and large radii were obtained from published data (Ahuja et al., 2021). Details of the calculations can be found in the 'comment' column of Supplementary Table S1.5, line #26.

### 5.6. Calculation of surface area for vessel cells

Vessel cells were modelled as cylinders, and xylem tracheids – devoid of cell walls at their extremities – were modelled as hollow

cylinders. Cell surface areas were calculated using cell lengths (see Supplementary Table S1.2, lines #27 to #31) and radii, calculated from the section area (see Supplementary Table S1.5, lines #95 to #99), assuming circular cell perimeters (see details in the 'comment' column in Supplementary Table S1.5, lines #109 to #113).

### 5.7. Calculation of cell wall volumes

In the absence of data from the literature for mesophyll and epidermis cells, we measured cell wall thickness by cryofracture CryoSEM, using plants grown in long-day conditions (stage 3.5, according to Boyes et al., 2001). Samples were prepared as described in Wightman et al. (2017) with the following modifications: The fracture was sputter coated with 5 nm Gold/Palladium and imaged with the SE detector at 6 kV gun voltage and 16 pA I probe size. The averaged cell wall thickness was 150 nm for epidermal pavement cells and mesophyll cells (see Supplementary Figure S2). We assumed the same thickness for bundle sheath cells (see Wei et al., 2021). Values for the other cells and the references are listed in Supplementary Table S1, sheet S1.5, lines #11 to #19.

To estimate the cell wall volume for each type of cell, the internal surface area of the cells was multiplied by the cell wall thickness (see Supplementary Table S1.5, lines #139 to #149 for individual cells). The proportion of cell wall relative to internal (i.e., 'aqueous') cell volume is presented in Supplementary Table S1.5, lines #152 to #162. The cell wall volume at the leaf level can be found in Supplementary Table S1.5, lines #165 to #175.

## 6. Procedure used to convert metabolite abundance in various units into concentrations in μM

The details of the procedure used to convert metabolite abundance values as found in the original studies into a single harmonised unit (concentrations in μM) are given in Supplementary Method 1, with the corresponding Python script and data used in Supplementary Table S4, sheets S4.1–S4.4. Input data (Supplementary Table S3) can be downloaded from the ChloroKB website by clicking on the download icon at the top of the home page. From the new page, item 4, users can download a CSV file containing the quantitative data present in ChloroKB. The calculation assumes that the volumes of cells and subcellular compartments remain the same across conditions and studies. The results of the data treatment using ChloroKB quantitative export are displayed in Supplementary Table S5, with the concentrations in μM displayed in the last columns in this table. The name of the metabolites displayed in the first column is duplicated in the final column to facilitate reading and use. Some metabolites are present under different isomeric forms in spontaneous equilibrium *in vivo* (e.g., α-glucose and β-glucose). The concentration indicated in Supplementary Table S5 for such isomers represents the sum of the concentrations of the different isomeric forms. The abundance of each specific isomer in the cellular environment can be calculated by applying the equilibrium constant.

### Acknowledgements

We thank Nathalie Wuyts for the image stacks used in Figure 1, and Michèle Crèvecoeur for kindly providing us with scanning electron microscopy pictures of an Arabidopsis trichome (see https://www.unige.ch/sciences/biologie/bioveg/crevecoeur/). We thank Ray Wightman for CryoSEM with cryofracture analyses (The Microscopy Core Facility at the Sainsbury Laboratory, University of Cambridge, supported by the Gatsby Charitable Foundation) and Zoe Nahas for growing the plants. We thank Elisa Dell'Aglio for critical reading of the manuscript and TWS Editing for advice on English usage. We thank Célian Curien for his help with the graphical abstract.

**Competing interest.** The authors declare they have no competing interests.

**Data availability statement.** Data used in this work are either referenced in the References section (main text) or in Supplementary Table S1 (column entitled 'references' under the format first author last name_year of publication_PMID). Metabolite quantitative data are available in Supplementary Table S3 (Excel format) and from ChloroKB website (http://chlorokb.fr, download section, item 4) (csv format). The python code used to convert original quantitative data into data expressed in a single unit and input tables are available on zenodo (https://doi.org/10.5281/zenodo.10410541; see also Supplementary Method 1 and the input tables provided in Supplementary Table S4, sheets S4.1–S4.4).

**Supplementary material.** The supplementary material for this article can be found at http://doi.org/10.1017/qpb.2024.1.

**Author contributions.** D.T. and G.C. collected data, organised them and carried out calculations. C.D.-T., E.N.S. and P.G. contributed to data formatting and python script writing. C.U. performed 3D reconstruction. D.T., D.F., D.V., E.N.S., G.F., Y.V. and G.C. wrote the manuscript.

**Funding statement.** G.C., G.F., D.T. and E.N.S. acknowledge funds received from the European Union H2020 Project 862087-GAIN4CROPS. This project received funding from the European Research Council: ERC Chloro-mito (Grant No. 833184) to G.C. and G.F. G.F. and C.U. acknowledge funds received from the European Union HORIZON programme, project 101099192-PLANKT-ON.

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
