## [Reviewer Report]

Review of “The Arabidopsis Leaf Quantitative Atlas: a cellular and subcellular mapping through unified data integration”

Tolleter and coworkers present a framework, essentially an Excel worksheet, as a tool to integrate and share quantitative structural and physiological data for plant structures. The framework is applied to the concrete case of the so-called “Leaf 6” of Arabidopsis. The work is cast as a step “to build a quantitative plant leaf atlas.”

The paper is clearly presented and is obviously very timely given the rapid growth in research data. In fact, one could argue that the plant community and the biology community at large have done too little to develop platforms that can host and integrate datasets so as to make them accessible to a broad range of researchers. This work is certainly a step in the right direction. Having said that, I do have a few suggestions for improvements.

1) As far as I can tell, the authors have not consulted or tried to follow some pre-established bioinformatics/data science protocols to develop their quantitative atlas. Am I not an expert in the field so I cannot point to a specific established protocol that would be optimal for this dataset but I would be surprised if none existed. At any rate, the introduction of the paper should at the very least include a review of the “dos and don’ts” when setting up a complex databank. One obvious issue with an Excel worksheet used in this work is that it is not centralized, nor curated. If the “Leaf Quantitative Atlas” were to be adopted by several groups, the dataset would quickly diverge unless a mechanism is established to maintain a consolidated Excel file.

2) The structural data regarding the cell types are all reported without reference to where the cells are found in the leaf. For example, many of the metrics have to do with the number of cells of given types and their geometrical properties (volume, etc.). Given this, segmented images and stacks would be the best way to store the data so as to preserve all of the relevant geometrical information. In fact, without the use of some type of image, the word “atlas” in the title seems misplaced. Instead of images, the integration of the information in this work is done with “numbers” in the Excel file itself. It would be good to discuss, in general terms, the “form” that ought to take the databank to achieve its goal. This could be accompanied with a detailed statement of how the databank is supposed to be used concretely.

3) I also have several minor issues about the metrics used by the authors. (a) The authors have decided to include direct measurements and derived measurements in the Excel file. Taking some of the macroscopic measurements as examples, some potential direct measurements are: the leaf surface area and the leaf dry weight. One derived metric is the “specific leaf area per unit leaf dry weight”. Since the latter metric can be derived from the direct measurements of leaf area and leaf dry weight, it is not clear to me why it should be included in the Table. More generally, what was the logic behind the selection of the metrics to include in the Table? (b) I was somewhat baffled to find that the leaf dry weight was inferred from other numbers instead of directly measured. Presumably, this is explained by the fact that the authors are using the data published in Wuyts et al. 2012 to fill much of the table. Hopefully, in due time, the metrics that are accessible to direct measurement will all have been measured directly. (c) Some of the metrics are accompanied with their experimental error. I could not find any information about how it was calculated. Is it the standard deviation or a more concrete error on the measurement calculated from the experimental protocol? If it is the SD, what are the sample sizes for the different values? The authors could also consider the simple rules of error propagation to put error bounds for ALL of the metrics. (d) Along a similar vein, the number of decimals in the values reported in Table 1 does not match the experimental error. For example, the leaf fresh weight is reported as 22.3 mg with experimental error 4.46 mg. It is not clear why the experimental error would be known with a precision of two decimal places (0,01 mg) while the reported value for the metric is precise to only one decimal place (0.1 mg). Also, if this error is rally related to the precision of the measurement, then the standard way to report the value is 22 ± 4 mg (i.e., the number of decimales in the measurement are adjusted to the precision of the experimental protocol and the precision itself has only ONE significant figure). Finally, some precision should also be stated for the cell numbers in Table 2 and Fig. 2. Are there really 104 375 xylem parenchyma cells? Surely, this number is not exact up to 1 cell.

4) One minor point, it would be good to say in Fig 1 which cell types are labelled in the figure and which are not so that the reader does not waste time looking for them.

---

## [Reviewer Report]

Review of MSC titled “The Arabidopsis Leaf Quantitative Atlas: a cellular and subcellular mapping through unified data integration” by Tolleter et al. submitted to Quantitative Plant Biology as original research article (QPB-23-0020)

This MSC describes a quantitative tool and a corresponding database dedicated to numerous traits of arabidopsis leaf (a nearly fully expanded rosette leaf #6, used as a model in wide spectrum of investigations). The addressed traits and parameters range from an organ to subcellular scale. The Leaf Quantitative Atlas includes datasets from literature that were carefully unified by the Authors, while methods, equations and sources are explained in an exhaustive way. Formulas embedded in Supplementary tables will allow readers to add new data to the existing dataset. This outstandingly comprehensive approach to gather and interpret quantitative data on the plant organ will in my opinion be a unique tool for researchers interested in multiscale modelling of a plant leaf and biochemical analyses of leaf function at the organ scale.

I have a few minor comments on this submission:

1. In line 166 please replace col-0 by Col-0

2. Lines 205-207: The explanation for vacuole volume is not clear (I understand what the Authors mean but this sentence needs to be improved)

3. Section titled “Calculation of membrane surfaces and cell wall volume.”: please consider changing the order of the second and third paragraph

4. Please unify figure reference – it has to be the same in Figs and the text in terms of small or capital letters, e.g. either 1A or 1a

5. Figure 1, panel c: please add the label “Abaxial epidermis” as in the case of other tissues

6. Figure S1 legend and other places in the text: please consider replacing the term “border” by “outline”

7. Figure S2 legend: please consider replacing “mesophyll / epidermal cells cell walls” by “mesophyll / epidermal cell walls” or “walls of mesophyll / epidermal cells“

8. Table S2: why are cambial cells referred to in the case of the still expanding leaf?

9. Supplemental_methods_1: Please remove a comment to the title (in French)

10. Will the python script be available to readers?

---

## [Editor Report]

Thanks for your submission to QPB and apologies for the delay with this review.

Both reviewers are supportive of the manuscript, raising a series of points to address. In particular reviewer 1 outlines areas to address before resubmitting a revised manuscript.

---

## [Reviewer Report]

I have been one of the reviewers of the initial submission of this manuscript and my opinion on the manuscript has already been positive. Thus now I can only state that the Authors have made all the suggested minor changes and the manuscript has been improved.